# Insights and Perspectives on the Role of Proteostasis and Heat Shock Proteins in Fungal Infections

**DOI:** 10.3390/microorganisms11081878

**Published:** 2023-07-25

**Authors:** João Neves-da-Rocha, Maria J. Santos-Saboya, Marcos E. R. Lopes, Antonio Rossi, Nilce M. Martinez-Rossi

**Affiliations:** Department of Genetics, Ribeirão Preto Medical School, University of São Paulo, Ribeirão Preto 14049-900, SP, Brazil; majucb@usp.br (M.J.S.-S.); marcos_hrl@hotmail.com (M.E.R.L.); anrossi@usp.br (A.R.)

**Keywords:** pathogenic fungi, fungal infections, antifungal therapy, stress-response, proteostasis, heat shock proteins, HSPs, UPR, UPS, ERAD

## Abstract

Fungi are a diverse group of eukaryotic organisms that infect humans, animals, and plants. To successfully colonize their hosts, pathogenic fungi must continuously adapt to the host’s unique environment, e.g., changes in temperature, pH, and nutrient availability. Appropriate protein folding, assembly, and degradation are essential for maintaining cellular homeostasis and survival under stressful conditions. Therefore, the regulation of proteostasis is crucial for fungal pathogenesis. The heat shock response (HSR) is one of the most important cellular mechanisms for maintaining proteostasis. It is activated by various stresses and regulates the activity of heat shock proteins (HSPs). As molecular chaperones, HSPs participate in the proteostatic network to control cellular protein levels by affecting their conformation, location, and degradation. In recent years, a growing body of evidence has highlighted the crucial yet understudied role of stress response circuits in fungal infections. This review explores the role of protein homeostasis and HSPs in fungal pathogenicity, including their contributions to virulence and host–pathogen interactions, as well as the concerted effects between HSPs and the main proteostasis circuits in the cell. Furthermore, we discuss perspectives in the field and the potential for targeting the components of these circuits to develop novel antifungal therapies.

## 1. Introduction

Fungal pathogens are a diverse and heterogeneous group of microorganisms that engage in intricate co-evolutionary dynamics with their host species. Because of their potential for infection, fungal pathogens have significant medical and agricultural implications. Fungal infections are a growing concern in medicine, particularly in immunocompromised individuals. Fungal pathogens can cause diseases ranging from superficial skin infections to life-threatening systemic infections. In immunocompromised patients, infections caused by fungi belonging to the genera *Aspergillus*, *Candida*, and *Cryptococcus* can be particularly harmful [1]. *Candida auris*, a multidrug-resistant species, has been identified as a global health issue, with mortality rates of approximately 60% in some countries [2].

In agriculture, fungal diseases can have enormous economic impacts and cause severe losses [3]. The major contributors to decreased yield and quality are infections caused by species such as *Blumeria graminis*, *Botrytis cinerea*, *Fusarium graminearum*, *Magnaporthe oryzae*, and *Ustilago Maydis*. These effects are potentiated by the production of harmful mycotoxins that contaminate food and feed [4].

Pathogenic fungi rely on several processes to regulate protein homeostasis and maintain normal cellular functions, growth, and survival, particularly under stressful conditions related to the host environment. These include the protein folding and degradation pathways [5]. For example, the endoplasmic reticulum-associated degradation (ERAD) pathway degrades misfolded proteins in the endoplasmic reticulum (ER). This process is essential for maintaining ER homeostasis [6]. Similarly, the ubiquitin–proteasome system (UPS) is a component of the intracellular protein degradation mechanism that degrades short-lived and misfolded proteins [7].

In addition to these mechanisms, the heat shock response (HSR) is also a major regulator of fungal proteostasis. HSR is mediated by heat shock proteins (HSPs), a group of proteins that act as molecular chaperones to prevent protein misfolding and aggregation [8]. HSPs are upregulated in response to various stressors, including heat shock, oxidative stress, nutrient deprivation, and antifungal drugs [9,10,11]. HSP regulation is mediated by heat shock factors (HSF), transcription factors that bind to heat shock elements (HSEs) in the promoter regions of HSP genes [12,13].

Given the rising prevalence of fungal infections, there is an urgent need to understand the biology of pathogens and their interactions with hosts. An integrated understanding of the molecular mechanisms that participate in fungal pathogenesis, such as proteostasis circuits, is necessary for developing effective treatments and management strategies for these agents in medical and agricultural contexts.

## 2. Proteostasis Circuits and Pathogenesis

### 2.1. Proteostasis during Infection

Fungal pathogens must quickly adapt to the challenging conditions of their hosts [14]. During infection, pathogens encounter numerous stressors that trigger protein misfolding and aggregation. Therefore, fungal pathogens rely on proteostatic regulation to maintain cellular homeostasis, enabling their survival and virulence under adverse host conditions [5]. A growing body of evidence has highlighted the importance of proteostatic circuits in fungal pathogenesis. Several studies have shown that the deletion of genes involved in these circuits, such as chaperones or proteases, reduces virulence. In *Cryptococcus neoformans*, deletion of the chaperone Hsp70 results in attenuated infection [15]. Similarly, the genetic repression of the chaperone Hsp90 affects several aspects of resistance, growth, and survival in *Aspergillus fumigatus* [16].

Fungal pathogens are exposed to high temperatures and pH shifts during infection. Increased body temperature is an important defense mechanism that assists host immune function and impairs pathogen proliferation in animal and human infections. Under these conditions, fungal HSR are highly integrated into other stress-responsive signaling pathways [17,18,19]. HSPs are expressed in *C. albicans* in response to acid stress in vaginal candidiasis. The overexpression of HSPs, particularly Hsp90 and Hsp70, aids in the correct folding and trafficking of virulence factors as well as *C. albicans* survival under acidic conditions [20,21,22]. Similarly, in *C. neoformans*, the HSR is triggered in response to alkaline stress, which occurs during lung and central nervous system infections. HSP regulation is also important for the survival of *C. neoformans* under alkaline conditions [23,24,25].

In addition to HSR, fungal pathogens have developed mechanisms to cope with changes in pH during infection. For example, in *Aspergillus* spp., the expression of an acid protease called aspergillopepsin A (PepA) is required for virulence [26,27,28]. *A. nidulans* can sense and respond to pH changes in the environment through a signaling cascade consisting of six genes (*palA*, *palB*, *palC*, *palF*, *palH*, and *palI*) whose products perceive alkalinization and promote the proteolytic processing of the PacC/RIM101 factor [29]. The pH-dependent cleavage of this transcription factor produces an active isoform that translocates into the nucleus and regulates the expression of genes involved in pH homeostasis and virulence. This system is conserved in fungi and works by activating genes that express products with optimal activity at alkaline pH and repressing acidic ones [30].

Pathogens must adapt to nutrient deprivation during infection. Nutrient availability is often limited in the host environment, affecting the structural and functional components of the cell and leading to impaired metabolism [31,32]. Fungal pathogens have developed mechanisms to cope with nutrient deprivation, such as the induction of autophagy, which allows for the recycling of intracellular components to maintain cellular homeostasis [33,34,35,36]. Autophagy is closely related to the expression of HSPs such as Hsp70 and Hsp90 [5,37]. These results highlight the importance of proteostasis in nutrient deprivation adaptation and the regulation of autophagy in fungal pathogenicity.

Additionally, in *Neurospora crassa*, the pathway responsible for inorganic phosphate acquisition is activated under scarce conditions, promoting the transcription of Pi-repressible structural genes. The signaling cascade is formed by the NUC-2 protein, a membrane component that senses changes in Pi levels and transmits a molecular signal to the PREG-PGOV complex. When deactivated, this cyclin-CDK complex allows for the translocation of the NUC-1 regulator into the nucleus [38,39]. This hierarchical regulation removes the repression of gene-encoding Pi transporters and enzymes, such as nucleases and pH-dependent phosphatases [40].

The presence of Hsf1, PacC, and NUC-1 consensus sequences in the promoter regions of HSPs and other related genes, including these transcription factors, provides unambiguous evidence for the molecular interactions of these pathways [41]. These findings indicate that fungal pathogens have highly sophisticated mechanisms for detecting and responding to environmental fluctuations. These circuits are closely linked to the biological aspects of interactions with the host. Fungal pathogens regulate the secretion of hydrolytic enzymes and other chemicals, including virulence factors, by monitoring environmental conditions. This mechanism leads to optimal metabolic capacity and acquisition for nutrient usage [42,43].

### 2.2. HSPs and Host–Pathogen Interactions

HSPs have been implicated in the adaptation of fungi to their host environment. For example, *A. fumigatus* has the unique ability to adapt to the host environment by modifying its cell wall composition and secreting different virulence factors. The upregulation of HSPs in *A. fumigatus* plays a critical role in these adaptive processes by promoting protein folding and trafficking and by modulating the expression of virulence factors [44,45,46,47,48,49,50]. During invasive infections, *A. fumigatus* encounters various host defense mechanisms, including reactive oxygen species (ROS) production and iron limitation [51,52,53,54,55]. In *A. fumigatus*, HSR is controlled by the transcription factor HsfA, which is essential for virulence, cell wall integrity, and sensitivity to stressors [56].

Oxidative stress is a common stressor experienced by fungal pathogens during infections. ROS produced during interactions with host compounds or immune cells is a potent antifungal defense mechanism [57,58]. To withstand oxidative stress, fungal pathogens must possess effective detoxification mechanisms such as synthesizing antioxidants and chaperones [10,59]. Indeed, several studies show that oxidative stress poses challenges to protein folding and stability, which recruit the function of heat shock proteins such as Hsp70 and Hsp90 [10,55,60,61].

Proteostasis also plays a role in the response of fungal pathogens to immune cells, thereby affecting the course of infection. HSPs interact with several components of the host immune system, including antigen-presenting T cells. Hsp70, for instance, is crucial for virulence and host–pathogen communication. In *C. albicans*, Hsp70 family members, in addition to their roles in the stress response and chaperoning of newly translated proteins, participate in many infection-related processes. This chaperone, which is present in the cell wall, plays an important role in secretory pathways and is found in extracellular vesicles [62,63,64]. These findings highlight the importance of Hsp70 in the virulence and protection of pathogens from host immune defense mechanisms (see the Section 3.1).

Thus, HSPs are essential for the survival and virulence of pathogens in host interactions. They may also play a role in the evasion of the host immune system. Hsp60, for instance, reportedly acts as an immunodominant antigen, triggering humoral and cellular responses that activate regulatory T cells by interacting with innate immune system components [65]. In *C. neoformans*, Hsp70 induces host macrophage M2 polarization, which helps colonize the respiratory tract [15]. Hsp90 is found on the cell surface of *Paracoccidioides lutzii* and *P. brasiliensis* and has a highly immunogenic profile that may play a role in pathogen identification by the innate immune system [66]. Thus, the complex interaction between fungal HSPs and the host immune system may modulate the activation and signaling pathways mediated by immune cells, leading to changes in the host response to the pathogen.

### 2.3. Protein Degradation and The Unfolded Protein Response

In eukaryotic cells, most secreted transmembrane proteins are folded and mature within the lumen of the endoplasmic reticulum (ER) [67]. This organelle plays a significant role in protein synthesis, folding, modification, and secretion [68]. Variations in the number of proteins in the ER lumen occur because of external and internal stimuli, such as environmental stress, cell differentiation, the physiological state of the cell, and nutrient starvation [67]. The accumulation of toxic unfolded or misfolded proteins in the ER triggers ER stress and activates the unfolded protein response (UPR) pathway [69].

One component of the UPR pathway is the transmembrane stress sensor Ire1/IreA (Ser/Thr kinase), which has an endonuclease (RNAse) domain that is activated under ER stress caused by the accumulation of misfolded proteins [70,71]. The chaperone protein commonly known as BiP (a member of the Hsp70 family) downregulates the UPR pathway by binding to the N-terminal region of Ire1 under normal conditions. Under stress, the BiP protein dissociates, resulting in Ire1 dimerization followed by binding to misfolded proteins through the central stress-sensing region in the sensor domain [72,73]. The RNase domain of Ire1 acts on the mRNA of the *hac*1/*hac*A gene, the central regulatory transcription factor in the UPR pathway [70,74]. Under nitrogen-sufficient conditions, the endonuclease domain removes an inhibitory intron from the *hac*1 mRNA and promotes the efficient translation of the Hac1p bZIP transcription factor [75]. Hac1p translocates to the nucleus and orchestrates transcriptional changes in various genes, such as chaperones, foldases, post-translational modification enzymes, and proteolytic proteins. Thus, UPR activation results in the upregulation of genes that support ER function in rebalance proteostasis [72,76].

Another important mechanism that supports the control of cellular stress and cytotoxicity caused by protein accumulation is the ERAD pathway, which mediates the degradation of misfolded or functional proteins with the assistance of the UPS [77]. The resident protein calreticulin and the integral membrane chaperone calnexin were the first to assist in ER protein quality control and ensure that only correctly folded and assembled proteins proceed further along the secretory pathway. Proteins that repeatedly fail to achieve their native conformation are retained in the ER and targeted for ERAD by demannosylation. These abnormal proteins are retranslocated to the cytoplasm for degradation by the UPS [78,79,80].

Given that pathogens must produce extracellular hydrolytic enzymes, which are necessary for nutrient acquisition and host invasion, a close relationship between HSPs and the ERAD, UPS, and UPR pathways has been observed in the context of fungal virulence (Figure 1). In *A. fumigatus*, the Δ*hac*A mutant is unable to activate the UPR pathway in response to ER stress, has a reduced capacity for protease secretion, and shows impaired growth when challenged to assimilate nutrients from complex substrates. In addition, the mutant was more susceptible to antifungal agents that disrupt the membrane or cell walls [76]. Similarly, in *Trichophyton rubrum*, virulence attributes such as keratinolytic capacity, thermotolerance, the growth of host molecules, and susceptibility to antifungals were affected in the Δ*hac*A mutant [74,81]. Regarding *C. albicans*, the mutation of this gene results in greater susceptibility to ER stressors and aberrant morphology and growth [82].

Greater susceptibility to stressors was also observed in ire1 mutant strains of *Kluyveromyces lactis* [72] and *Saccharomyces cerevisiae* [83]. Regarding *C. neoformans*, studies involving Bip chaperone and ire1 mutants have demonstrated that deficient UPR function impairs virulence attributes [68,73]. This evidence reinforces the importance of the UPR pathway in responding and adapting to environmental stress, suggesting that this pathway is a potential target against fungal pathogens.

## 3. Heat Shock Proteins

### 3.1. HSP Paralogs

Paralogs are genes related to each other by duplication, usually holding identical or similar functions but subjected to distinct selective pressures [84,85]. Fungi have several paralogs as functional alternatives to maintain homeostasis. These genes can be differentially expressed depending on the conditions to which they are subjected. For example, some genes of the Hsp70 family are expressed at ambient temperatures and are not induced by heat shock. In contrast, others exhibit the opposite behavior, even though they belong to the same gene family [86]. In *S. cerevisiae*, two Hsp90 paralogs, Hsp82 and Hsc82, show 97% identity in their amino acid sequences and differences in gene expression [23].

HSP paralogs play important roles in pathogenicity and virulence. The SSA1 and SSA2 Hsp70 paralogs in *C. albicans* have specialized roles because they are present in the cell wall. The *C. albicans* SSA1 mutant exhibits attenuated virulence in murine models of disseminated and oropharyngeal candidiasis [87]. SSA1 and SSA2 can also bind to human salivary histatin 5, increasing yeast survival in the presence of this defense peptide with potent antifungal activity [23,88]. The keratinolytic activity of dermatophyte fungi and their potential to infect biological tissues are guided by the upregulation of Hsp60, Hsp70, and Hsp90 transcripts. During a temperature increase, *T. mentagrophytes* showed a high expression of Hsp60 [65]. Several HSP paralogs have been described and characterized in dermatophytes over the years [41,89]. In the plant pathogen *F. oxysporum*, seven HSPs with sizes ranging from 18 to 95 kDa (18, 35, 70, 74, 80, 83, and 95 kDa) were identified along with the *STI35* gene in response to a sudden increase in temperature to 45 °C [23].

The genus *Trichoderma*, which is composed of fungi capable of parasitizing other fungi, also contains several HSP paralogs in its genome. *T. asperellum* has four representatives of the sHSP family (small HSPs), twelve of the Hsp70 family, three of Hsp104, and only one of Hsp90. In addition, the proteins encoded by TaHsp26a, TaHsp26b, TaHsp26c, TaHsp70a, TaHsp70b, TaHsp70c, TaHsp90, TaHsp104a, and TaHsp104b showed homology with other *Trichoderma* species, such as *T. atroviride*, *T. harzianum*, *T. reesei,* and *T. virens* [90,91]. *Trichoderma* species are used as biological controls for the phytofungi *Sclerotinia sclerotiorum* and *F. oxysporum*, and paralogs are likely to be important during the mycoparasitism of *S. sclerotiorum* [91].

In the dimorphic fungus *P. brasiliensis*, high expression of Hsp70 paralogs is fundamental for stabilizing the splicing machinery in the yeast form. This stabilization results in the transcription of other genes expressed during the growth of *P. brasiliensis* within the host, consequently reinforcing its pathogenicity [86]. HSP paralogs and their counterparts have been identified in several phytopathogens. *Bremia lactucae* contains three Hsp70 paralogs. *Phytophthora infestans* contains Hsp70 paralogs that are also found in *P. parasitica*: P1H2CY485, P1H4CY390, and P2H3CY300 [92]. The oomycete phytopathogen *Achlya ambisexualis* uses the expression of Hsp70 and Hsp90 to generate ramifications in the hyphae, thereby increasing its pathogenicity. This fungus contained two virtually identical Hsp90 transcripts. In the presence of the anteridiol pheromone produced by *Achlya* species, there is a significant presence of Hsp90 transcripts and three Hsp70 paralogs [93].

Overall, paralogs play a critical role in the adaptation to different environmental conditions and pathogenicity. In some fungi, the upregulation of HSP paralogs can increase survival and virulence. Conversely, in others, it is fundamental for the transcription of genes expressed during growth and even for interacting with hosts. Understanding the adaptive roles of HSP paralogs will lead to a better understanding of fungal pathogenesis and strategies to cope with proteostatic problems during disease.

### 3.2. Small HSPs

Small Heat Shock Proteins (sHSPs) are ATP-independent chaperones that delay the onset of protein misfolding and the initiation of aggregation [94] and participate in various subcellular compartments [94,95,96]. These proteins usually range in size from 12 to 43 kD with a dynamic ability to form oligomers [94]. The available amino acid sequences of these proteins demonstrate the presence of three main essential domains: the amino-terminal region (NTR), α-crystallin domain (ACD), and the carboxy-terminal region (CTR) [94,96]. The α-crystalline domain is conserved and comprises seven or eight-antiparallel chains, forming a β-sandwich responsible for dimer formation, and the amino-terminal extension is thought to influence the higher-order oligomerization dynamics of subunits and chaperoning. In contrast, the highly flexible, charged carboxy-terminal extension stabilizes the oligomers while mediating the chaperone activity of sHSPs [94,97].

sHSPs provide protection against many stressors, and their absence in cells simultaneously affects many processes involving target proteins (Figure 2) [98]. sHSPs have been described in various pathogens, including *Aspergillus*, *Botrytis*, *Fusarium*, *Magnaporthe*, *Neurospora*, *Penicillium*, and *Trichophyton* species [99]. In addition to growth regulation and stress tolerance, the fungus *U. maydis* contains Hsp12, Hsp20, and Hsp30, which are important components of pathogenicity [100]. In *A. nidulans*, these sHSPs are involved in various conditions such as thermal/cold, osmotic, and oxidative stress [99]. In *C. gattii*, Hsp12 is also involved in virulence and phagocytosis, demonstrating its importance during infection [101].

In *Schizosaccharomyces pombe*, heat shock treatment increased the expression of *hsp16* by 64-fold in wild-type cells and 141-fold in cdc22-M45 (ribonucleotide reductase) mutant cells [102]. In addition to HSR, the expression of Hsp16 is mediated by the Spc1 MAPK signaling pathway through the ATF1 transcription factor. Nucleotide depletion or DNA damage is sufficient to activate *hsp16* expression through ATF1, suggesting a role for small HSPs in the stress response following nucleotide depletion or DNA damage [102]. Similarly, the same heat shock treatment induces both Hsp16 and Hsp15.8 in *S. pombe*, and sHSP-null mutants display significant heat sensitivity, indicating their importance in thermal stress management [103].

Efforts to elucidate the role of these proteins have shown that, among the sHSPs, Hsp30 is involved in the response to various stress conditions, such as temperature and pH fluctuations, osmotic pressure, and antifungals [104,105]. Hsp30 is an integral plasma membrane protein and a stress-inducible regulator of plasma membrane H+ ATPase. Hsp30 works by downregulating the activity of H+ ATPase during ATP depletion, thus playing a role in energy conservation [106]. Glucose depletion, ethanol stress, and acetic acid induce chaperone synthesis [107]. N-glycosylation, a widespread covalent modification of membrane and secretory proteins in eukaryotes, is induced at Asn2 of Hsp30 by severe heat shock, ethanol stress, and acetic acid stress. Mild heat shock and glucose depletion induce the expression (but not N-glycosylation) of Hsp30, indicating that N-glycosylation is dependent on temperature and environmental conditions [107]. Hydrostatic pressure also modulates Hsp30, and the loss of this chaperone impairs high-pressure growth [108].

In the human pathogen *T. rubrum*, the regulation of Hsp30 is associated with the growth of keratin, suggesting a possible role in the pathogenesis of dermatophyte infections [109,110,111]. Similarly, in *P. brasiliensis*, Hsp30 plays a role in the successful progression of the parasitic phase and is related to stress response and development [112]. In *N. crassa*, the expression of Hsp30 is detected during the transition from the mycelial to the yeast form [113]. In addition, the absence of this gene in *N. crassa* led to a greater susceptibility to heat shock under glucose restriction. Concomitantly, the enzyme glucose-phosphorylating hexokinase, which is recruited for glucose metabolism, was reduced by more than 35% in Hsp30 mutant cell extracts. Furthermore, K-crystallin- and Hsp30-enriched preparations protected the purified hexokinase from thermal inactivation in vitro. This indicates that Hsp30 directly stabilizes hexokinase during heat stress and contributes to the stress response of glucose metabolism [113]. In *C. glabrata*, Hsp30 helps to successfully colonize and adapt to high concentrations of acetic acid in the vaginal tract. Hsp30 contributes to a reduction in the accumulation of this toxic compound in *C. glabrata* cells [114]. In *Penicillium marneffei*, a dimorphic fungal pathogen that can cause disseminated mycosis, Hsp30 is upregulated in yeast cells grown at 37 °C compared to mycelial cells at 25 °C [115]. Heat shock also induces the expression of Hsp30 in *A. nidulans*. Interestingly, Hsp30 is preferentially produced in an acidic milieu, and its expression depends on the *pal*A+ genetic background, providing new insights into the connection between temperature- and pH-sensing networks [116].

Despite the fact that sHSPs were discovered a considerable amount of time ago, more efforts are required to explore their functionality during fungal infections.

### 3.3. Hsp60

Proteins of the Hsp60 family are multifunctional and participate in more than one independent biological process and distinct cellular compartments, particularly in the mitochondria [117,118,119]. The association between Hsp70 and Hsp60 homologs facilitates the folding and transport of proteins throughout the cell (Figure 3). Moreover, mitochondrial Hsp70 mediates Hsp60 biogenesis [120].

Hsp60 chaperonins are highly immunogenic proteins in dimorphic fungal pathogens such as *Histoplama capsulatum*, *P. brasiliensis*, and *P. lutzii* [117,118]. In *H. capsulatum*, it has been demonstrated that the fungus uses Hsp60 located in the cell wall as a ligand for CD11/CD18 in human macrophages. Cellular localization was confirmed using four experimental methods, including immunoelectron microscopy and flow cytometry [117]. In the dimorphic fungus *P. brasiliensis*, Hsp60 expression is increased in both hyphal and yeast forms, in addition to transcription in the transitional morphology of this fungus. The presence of Hsp60 in the cell wall of *P. brasiliensis*, mainly in the yeast form, highlights the importance of this chaperonin in its pathogenicity [118]. In *Sporothrix schenckii*, a dimorphic fungus that infects animals and humans, Hsp60 functions as an adhesin, allowing its extracellular matrix to bind to host components and trigger the immune system [121].

### 3.4. Hsp40/Hsp70

The 70-kDa heat shock protein family Hsp70 is highly conserved and centralized in proteostatic circuits [122]. Important functions of Hsp70 include controlling the activity of regulatory proteins, folding and assembling newly synthesized proteins, refolding denatured or aggregated proteins, and translocating proteins to different organelles [123]. Thus, Hsp70 functions as a chaperone and is functionally centralized to guarantee cellular homeostasis during proteotoxic stress (Figure 4) [124]. The role of Hsp70 is linked to joint actions with its co-chaperone Hsp40 and other HSPs such as Hsp90 and Hsp104 [104,125]. Despite their highly conserved functions, the Hsp70 family members have a high degree of specificity [124,125]. Three main factors differentiate Hsp70 paralogs: (1) substrate recognition and allosteric regulation; (2) specificities guided by the target site in cell organelles, such as guiding a newly synthesized protein by the ribosomes to the nucleus; and (3) binding with the J domains of Hsp40 [126] and with NEFs (nucleotide exchange factors), which control the release of client proteins [124]. The recognition of misfolded proteins by Hsp40/Hsp70 occurs in hydrophobic regions that are exposed when proteins are denatured. In summary, Hsp40 functions as a linker between substrates, and Hsp70. Hsp40 recognizes and delivers suitable Hsp70 substrates by binding to its J-domain Hsp70 (Figure 5). Hsp40/Hsp70 interactions are fundamental to pathogenicity in pathogenic fungi [80,123].

In *A. fumigatus*, Hsp40/Hsp70 expression increases thermotolerance from 25 to 35 °C [127]. In *A. nidulans*, Hsp70 ensures fungal survival under changing pH [104]. Hsp70 also plays essential roles in germination, conidiation, and sporulation in *N. crassa*, in addition to playing a role in the response to oxidative stress [128]. Furthermore, in *Histoplasma* spp., Hsp70 has been associated with stress adaptation early during the mycelium-to-yeast transition [119]. In the pathogenic yeast *C. neoformans*, Hsp70 participates in processes that allow the fungus to interact with the alveolar epithelium or macrophages. Analyses of gene expression in *C. neoformans* cells found in the central nervous system of infected rabbits have revealed the importance of upregulating Hsp12, Hsp60, Hsp70, and Hsp90 families for successful infection [23,129]. The plant pathogen *M. oryzae*, in turn, has a fascinating and newly discovered adaptive mechanism. This species has three cytosolic forms of HSPs: MoSsb1, which is homologous to a typical Hsp70; MoSsz1, which is an atypical Hsp70; and MoZuo1, an Hsp40 [130]. The complex formed by these three proteins contributes to the development, conidiation, and pathogenicity of this phytofungus through interactions between HSPs and the MAPK pathway. This pathway is related to cell-wall integrity in *M. oryzae* [128,130].

Within the mitochondria, the folding of imported proteins to meet their target sites is mediated by the ATPase activity of Ssc1, a local Hsp70. Mdj1, an Hsp40 homolog, is essential for mitochondrial biogenesis and function in yeast. Lon, which works with Mdj1, is a conserved proteinase that participates in the disaggregation and cutting of misfolded proteins, whereas Mdj1 is essential for substrate degradation [131]. In dimorphic pathogenic fungi such as *P. brasiliensis*, *P. lutzii*, and *H. capsulatum*, a pattern similar to that previously observed in *A. nidulans* and *A. fumigatus* occurs; Mdj1 is present in the cell wall, and its disposition varies according to temperature. This suggests that mitochondrial Hsp40, like Hsp60, could also play a role in host–pathogen interactions, growth, and yeast–mycelium transition [131]. Furthermore, the organization of the LON/MDJ1 locus is highly conserved among pathogenic *Eurotiomycetes* species [132], which may indicate a greater role for HSPs in virulence.

Finally, in dermatophytes, the amino acid sequence of one of the Hsp70 genes in *T. rubrum* shows high identity with Hsp70 genes in other pathogenic fungi such as *N. crassa, C. herbarum*, *C. albicans*, and even human Hsp70 [133]. Northern blot analyses have indicated that the Hsp70 gene is constitutively expressed at low levels [133] but upregulated under stress [89,134]. In the presence of antifungals such as Amphotericin B, the induction of genes that encode stress-related proteins occurs, including proteins from the Hsp70 and Hsp104 families [135].

### 3.5. Hsp90

Heat shock protein 90 (Hsp90) folds numerous target proteins that participate in a wide range of cellular processes. In fungi, Hsp90 has been identified as a key player in the regulation of virulence, because it is involved in the maturation and stabilization of the proteins required for pathogenicity and resistance [136]. Hsp90 is a dimeric protein consisting of three main domains: N-terminal, middle, and C-terminal domains. The middle and C-terminal domains are important for the binding and folding of client proteins, whereas the N-terminal domain is responsible for ATP binding and hydrolysis [80,137]. Hsp90 functions with co-chaperones, such as Hop/Sti1 (Hsp organizing proteins), CHIP (ubiquitin ligase), Cyp40, and Hsp40/Hsp70. These proteins promote the binding of client proteins and assist in controlling the ATPase activity of Hsp90 [80].

Hsp90 has been linked to fungal pathogenicity in numerous studies because the functions of different factors involved in resistance, morphogenesis, and virulence are tightly controlled by Hsp90. This chaperone regulates the cell cycle and the yeast-to-hyphae transition, which are crucial stages in the pathogenesis of *C. albicans* infections [138,139]. The Ras1 signaling system, which regulates hyphal growth, reproduction, and development at high temperatures, is dependent on Hsp90 in *C. albicans* [138,140]. Hsp90 interacts with distinct proteins in biofilms and planktonic cellular states [141]. Many of these client proteins are transcription factors crucial for the development of biofilms and the attachment of host cells [142,143]. In *H. capsulatum*, Hsp90 is important for virulence and survival during host–pathogen-associated stress [119]. Hs90 has been shown to regulate capsule induction and maintenance in *C. neoformans* and mediate virulence and resistance to antifungal drugs in an invertebrate model of infection [144,145]. In dermatophyte fungi, Hsp90 is essential for the successful colonization of keratinized tissues. In an ex vivo nail infection model, Hsp90 inhibition affected the ability of *T. rubrum* to degrade keratin [89] and reduced its growth and virulence [11]. Overall, Hsp90 articulates morphological and stress-associated programs that promote virulence in different fungal pathogens [48,146,147,148].

Hsp90 is present in high concentrations in the cytosol and can be further upregulated under stress conditions. Unlike Hsp70, Hsp90 is not responsible for binding to unfolded proteins but instead binds to a defined set of client proteins, such as kinases, cell surface receptors, and transcription factors. Thus, Hsp90 acts on these pre-folded proteins, assisting their stabilization and maintaining them in a functionally folded state, especially under stress conditions (Figure 6) [80].

An interesting feature of Hsp90 is its role as a molecular transistor. Hsp90 modulates the activity of kinases and transcription factors in response to environmental cues, tuning the activity of the cellular signaling network to subtle thermal fluctuations and proteotoxic stresses that influence Hsp90 availability [149]. This mechanism allows for the rapid and reversible modulation of protein activity and changes in cellular signaling. Hsp90 also acts as a capacitor in evolutionary timescales. Its buffering presence in the cell stabilizes proteins, allowing client kinases to accumulate point DNA mutations and maintain proper folding. Under stress, chaperone availability is reduced, and adaptive or deleterious effects of such mutations arise. This effect promotes differential evolution by harboring genetic variation [150].

In summary, Hsp90 plays a critical role in fungal pathogenicity by regulating the folding and stability of key virulence factors. Its ability to act as a molecular transistor also allows for rapid signaling changes that are essential for successful infection. Future research will likely continue to explore the role of Hsp90 and its cochaperones in fungal pathogenesis.

### 3.6. Hsp104 and Aggregation

Protein aggregation is a common phenomenon in cells under stress conditions. In particular, proteins with complex structures, prion-like domains, or intrinsically disordered regions are prone to misfolding under stressful conditions. Misfolded proteins tend to aggregate, forming highly ordered amyloid fibrils or amorphous aggregates [151,152,153]. One protein that plays a critical role in preventing and reversing protein aggregation in fungi is Hsp104 (Figure 7). Hsp104 is a member of the AAA+ superfamily of ATPases present in a wide range of organisms. This protein is characterized by a hexameric ring structure that uses energy from ATP hydrolysis to disassemble protein aggregates and prevent amyloid fibril formation [154]. The action of Hsp104 in protein disaggregation is complex and involves multiple steps. First, Hsp104 recognizes and binds to protein aggregates using its ATPase activity to unwind the aggregates into smaller oligomers. These oligomers are then processed by other chaperones such as Hsp70 and Hsp40 to refold them into their native conformation [154,155,156].

In the plant pathogen *M. oryzae*, Hsp104 is required for the normal development of the appressoria, a specialized structure that allows the pathogen to invade plant cells. In addition, functional crosstalk between Hsp104 and the autophagy machinery is observed under infection-related conditions. In many cases, aggregated proteins are spatially sequestered into quality control compartments, and the autophagy mechanism is required for the normal formation and compartmentalization of protein aggregates [157]. Hsp104 also plays a role in the virulence of *F. pseudograminearum,* and its deletion significantly impairs fungal growth, conidiation, and virulence in wheat and barley [158]. In clinical yeasts, proteins with the ability to aggregate in response to stress can assemble into a variety of states, including prions. Some fungal prions can have an adaptive role in phenotypic plasticity and stress response and can also be passed down through generations via Hsp104-dependent protein-based epigenetics [159].

In addition, protein aggregation is associated with biofilm formation, a common feature of infections with high mortality rates [160]. Fungal cells form organized communities called biofilms, which are encased in a protective extracellular matrix. This effect increases resistance to environmental stress and antifungal medications [161]. The formation of the extracellular matrix and preservation of biofilm structure are both thought to be significantly influenced by protein aggregation, and amyloid-forming proteins are present in biofilms [162]. Studies have shown that Hsp104 is required to establish and develop biofilms in *C. albicans*, likely due to its pivotal role in the biology and turnover of protein aggregates [163,164]. The precise mechanisms by which Hsp104 affects biofilm formation and fungal virulence are yet to be uncovered. In summary, Hsp104 plays a critical role in proteostasis and the maintenance of protein quality control in fungal pathogens. Its function as a disaggregase is essential for clearing toxic aggregates and restoring protein homeostasis under stressful conditions.

Table 1 reviews the main roles of proteostatic circuits and their importance for fungal pathogenicity. Information about drugs targeting components of these circuits is also provided, which can help guide further studies in the field.

## 4. Antifungal Therapy

HSPs, including Hsp70 and Hsp90, have been identified as potential drug targets in parasites such as *P. falciparum* and *A. terreus* [180,181]. Studies targeting the Hsp90–calcineurin pathway against several pathogenic fungal species have been performed using the calcineurin inhibitors Cyclosporine A and FK506 (tacrolimus), Hsp90 inhibitor Geldanamycin and its derivatives, and Trichostatin A, which inhibits Hsp90 function by inducing Hsp90 acetylation [16]. While Geldanamycin alone showed poor results against the majority of fungi, Trichostatin A exhibited antifungal activity and demonstrated effectiveness against resistant species such as triazole-resistant *A. ustus*, Amphotericin B-resistant *A. terreus*, and multidrug-resistant *Scedosporium prolificans*. Trichostatin A also synergized with Caspofungin against *A. ustus*, and Geldanamycin synergized against *Rhizopus* spp. Azoles exert their antifungal action by inhibiting the C14α demethylation of lanosterol, thus interfering with the synthesis of ergosterol. Amphotericin B acts by inserting itself into the fungal membrane in close association with ergosterol, forming porin channels that lead to loss of transmembrane potential. Conversely, Caspofungin inhibits the synthesis of β-glucan, an essential component of the fungal cell wall [182,183]. Although Trichostatin A appears to be a good candidate for antifungal therapy due to the low dosage required against resistant fungi, the short half-life of this compound is a significant limitation [16]. FK506 has an additive effect on Hsp90 repression via Geldanamycin; however, the immunosuppressive properties of this calcineurin inhibitor outweigh its antifungal action in vivo [16,184]. Cyclosporine A might be a more suitable inhibitor in this case since it has significant antifungal effects against *C. neoformans* and has demonstrated synergistic properties with Amphotericin B [185,186].

In *A. fumigatus*, Hsp90 inhibition with geldanamycin increases azole and echinocandin susceptibility [136,187,188]. Similarly, in *C. neoformans* and *C. gatii*, the inhibition of Hsp90 increases the susceptibility to Amphotericin B and Fluconazole [144]. Efungumab, a monoclonal antibody targeted against Hsp90, has been tested against *Candida* infections. In animal trials, synergy between Amphotericin B and Efungumab was demonstrated by an increase in negative biopsies of Fluconazole-resistant *C. albicans* (kidney), *C. krusei* (spleen), *C. glabrata* (spleen), and *C. parapsilosis* (liver and spleen) [189]. Efungumab also acts synergistically with Caspofungin, revealing the versatility of using HSPs inhibitors with commercial antifungal drugs [190,191]. In *T. rubrum*, the chemical inhibition of Hsp90 results in increased susceptibility to antifungal agents. The functional inhibition of ATP binding to Hsp90 disassembles the molecular complex between Hsp90 and target proteins and cochaperones, consequently abrogating drug resistance and increasing the efficacy of traditional antifungal drugs, such as Itraconazole and Micafungin [11].

Moreover, Hsp90 is a potentiator of evolutionary changes, facilitating the rapid emergence of drug resistance traits. By buffering genetic variation, Hsp90 provides a source of phenotypic plasticity that can rapidly produce novel phenotypes, including those resistant to antifungals [149,192]. Cowen and colleagues [136] investigated the role of Hsp90 in the development of drug resistance in *C. albicans* and *A. terreus*. The authors found that the inhibition of Hsp90 activity blocked the emergence of resistant phenotypes in both species. The study also showed that Hsp90 buffering capacity can alter mutation and recombination rates, leading to the rapid evolution of resistance in fungi.

The inhibition of Hsp70 can be achieved through various approaches, such as specific RNAi constructions or pharmacological substances like KNK-437, 2-phenylethynesulfonamide (2-PES), Pifithrin-μ, or Methylene Blue [80,181]. The inhibition of this central chaperone leads to the accumulation of aggregated and insoluble proteins, affecting protein quality control systems. The inhibition of Hsp70 in *A. terreus* also increases the effectiveness of Amphotericin B, demonstrating its involvement in antifungal resistance [181]. In *A. nidulans*, translocation of the AstA transporter into the mitochondria is related to survival under microaerophilic conditions, elevated temperatures, surface growth, biofilm formation, and penetration of the host tissue. Methylene Blue prevents AstA translocation into the mitochondria [193]. In addition, alternative splicing may occur in the transcripts of HSPs and act as an adaptive mechanism against stress in fungal cells. In dermatophytic fungi, this process is coordinated with RNA turnover. This enables the regulation of Hsp70, allowing fungi to survive under stressful conditions, including those caused by antifungal drugs such as Terbinafine and Undecanoic Acid [89].

The inhibition of Hsp90 often results in increased proteasomal degradation in cells. Arguably, the depletion of Hsp90 levels impairs client protein stabilization, leading to ubiquitination and proteasomal degradation. Because Hsp90 client proteins include transcription factors and kinases, the proteasomal degradation of these clients during Hsp90 suppression causes significant disruptions in numerous signal transduction pathways. In contrast, the silencing or impairment of Hsp70 reduces proteasomal activity, leading to the accumulation of misfolded, insoluble proteins because of their key position during protein refolding or degradation. The simultaneous inhibition of Hsp90 and Hsp70 enhances the cytotoxic effects of HSP inhibitors and promotes apoptosis [80].

In addition to other HSPs, the importance of Hsp104 in fungal pathogenicity has led to the development of Hsp104 inhibitors as potential antifungal agents. These inhibitors specifically target Hsp104 ATPase activity, disrupting its disaggregase activity [194]. Studies exploring Hsp104 inhibition are currently underway. Furthermore, Hsp60 has shown promising preliminary outcomes as a vaccine component targeting dermatophytes such as *T. mentagrophytes* [195]. Using Hsp60 as a vaccine component has yielded interesting results against *H. capsulatum* and P. brasiliensis [119,196].

In summary, although HSPs have shown potential as drug targets, their implementation requires careful consideration because of the similarity between fungal and host cell structures. The combination of HSP inhibitors with existing antifungal drugs holds promise for improving therapeutic outcomes. However, considering the global increase in fungal infections and emergence of resistant strains, further research is needed to understand these mechanisms and develop effective therapies against fungal infections.

## 5. Conclusions and Perspectives

Our understanding of host–pathogen interactions can be improved by clarifying the processes by which proteostasis contributes to fungal pathogenesis. Research on the interface between HSPs and host defense mechanisms, including immune responses in the context of infections, can lead to the identification of immunomodulatory targets. Therefore, studying proteostatic circuits in fungal infections holds great promise for enhancing our understanding of virulence strategies and developing novel therapeutic approaches to combat these pathogens. In Figure 8, we provide a schematic overview of the main proteostatic circuits discussed here.

HSP paralogs have evolved to fulfill specific roles and adapt to selective pressures. For example, investigating the functions of other Hsp70 paralogs could provide insights into substrate recognition, organelle targeting, and interactions with cochaperones. Further research on the specific roles of HSP paralogs should provide insights into fungal strategies for coping with proteostatic problems during infections. In addition, understanding how HSP paralogs function and are regulated in various species may offer new insights into the molecular mechanisms underlying fungal pathogenesis.

However, the precise functions and contributions of sHSPs to fungal infections remain largely unknown. Contemporary research has focused primarily on larger HSP, overlooking their unique features and potential significance. Further experimental studies on sHSPs are necessary to understand the proteostasis circuits and offer opportunities for targeted interventions. However, the implication of Hsp60 in fungal pathogenicity requires further investigation. The presence of Hsp60 in the cell walls of fungal pathogens indicates its importance in virulence, warranting additional investigations and consideration as a target for therapeutic interventions, including vaccines. Hsp70 and its cochaperone Hsp40 perform essential protein folding, assembly, and transport functions. Cellular homeostasis depends on the cooperation of other HSPs, including Hsp90 and Hsp104. The distinct roles of Hsp70 in key biological processes, coupled with the specificity of Hsp70 paralogs and their interactions with client proteins, highlight the potential use of this protein as a therapeutic target. In addition, Hsp90 clients include kinase proteins and transcription factors, which are central components of signaling pathways.

Thus, HSPs, including Hsp70 and Hsp90, have been identified as prospective targets and inhibitors of these chaperones and have been studied for their antifungal action. Combining Hsp70 and Hsp90 inhibitors or HSP inhibitors with currently available antifungals may enhance therapeutic outcomes and prevent drug resistance. It has been largely demonstrated that inhibiting Hsp90 increases the susceptibility of fungi to existing antifungal drugs. Hsp90 has been implicated in the emergence of drug resistance traits by buffering genetic variation and providing phenotypic plasticity [136]. Hsp90 inhibition as a treatment strategy might not only directly affect fungal virulence but also prevent the emergence of resistant phenotypes and enhance therapeutic outcomes.

Overcoming the challenges surrounding antifungal therapy is a compelling task as fungi are increasingly displaying drug resistance traits. Interest in the possibility of targeting HSPs has been stoked by the pressing need for novel therapeutic targets in the fight against fungal infections. However, given that HSPs are present in both fungi and hosts, there is a risk of unwanted side effects and an incomplete drug response. Creating inhibitors that specifically target fungal HSPs while reducing their off-target effects is challenging. Further research is needed to better understand the specificities of HSP function in fungal pathogenicity and to design safe and effective therapies that do not compromise human health.

Hsp104 plays a well-defined role in protein disaggregation and reactivation under stress conditions. This chaperone is mainly found in certain fungi and bacteria. Although plants, humans, and animals generally harbor other HSPs such as Hsp70 and Hsp90, they do not have a homologous counterpart to Hsp104 [197,198]. The absence of Hsp104 in these organisms, together with its unique properties and functions in fungal pathogens, makes it a promising target for developing antifungal strategies specifically tailored to combat pathogens. The absence of Hsp104 in animals and plants also suggests that it may have evolved as a specialized protein in fungi and bacteria to fulfill specific functions related to protein disaggregation and refolding. However, compared to other major HSPs, Hsp104 has yet to be extensively studied, and its functions and mechanisms need to be further explored in the context of fungal infections and antifungal therapy.

Protein aggregation is a common phenomenon in cells undergoing stress, and misfolded proteins have an innate tendency to form aggregates [199]. Fungal prions have been described in *S. cerevisiae* as a particular type of functional protein aggregate [200]. Yeast prions constitute an important adaptive mechanism for survival under stress conditions, and prion propagation requires Hsp104 disaggregase activity [201]. However, the role of prions in fungal pathogenesis, which constitutes a broad and open field of research, is yet to be elucidated. Furthermore, we believe that the understanding of fungal infections could greatly benefit from further research into in vivo protein aggregation.

## Figures and Tables

**Figure 1 microorganisms-11-01878-f001:**
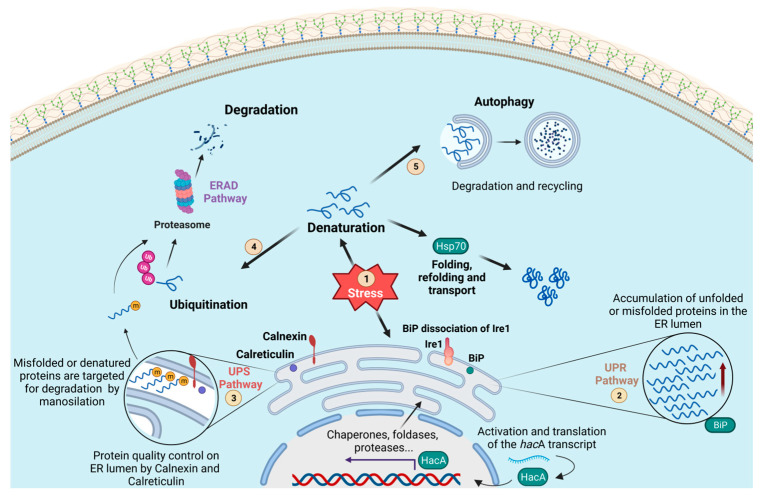
**Schematic representation of the processes for the maintenance of cellular proteostasis.** In this overview, the unfolded protein response (UPR) (2), ubiquitin–proteasome system (UPS) (3), endoplasmic reticulum-associated degradation (ERAD) (4), and the autophagy (5) pathways are represented. These systems are activated after some stress reaches the cell (1). The UPR pathway (2) is composed of the transmembrane stress sensor Ire1/IreA (Ser/Thr kinase) and the chaperone protein commonly known as BiP (a Hsp70 family member). Under stress conditions, the central regulatory transcription factor in the UPR pathway (Hac1/HacA) is activated by Ire1/IreA. It migrates to the nucleus, regulating the expression of genes that support endoplasmic reticulum proteostasis. The chaperones calnexin and calreticulin cooperate in maintaining the quality control of ER proteins. Proteins that fail to reach their proper conformation are then targeted by mannosylation and transported to the cytoplasm for further degradation by the UPS pathway (3). Misfolded proteins coming from the cytoplasm or arriving from other cellular compartments are also targeted by ubiquitination for recognition by the proteasome and are then degraded (ERAD 4). The process of autophagy (5) also operates in association with HSPs to maintain cellular proteostasis by recycling/degrading misfolded proteins in a lysosome-dependent manner.

**Figure 2 microorganisms-11-01878-f002:**
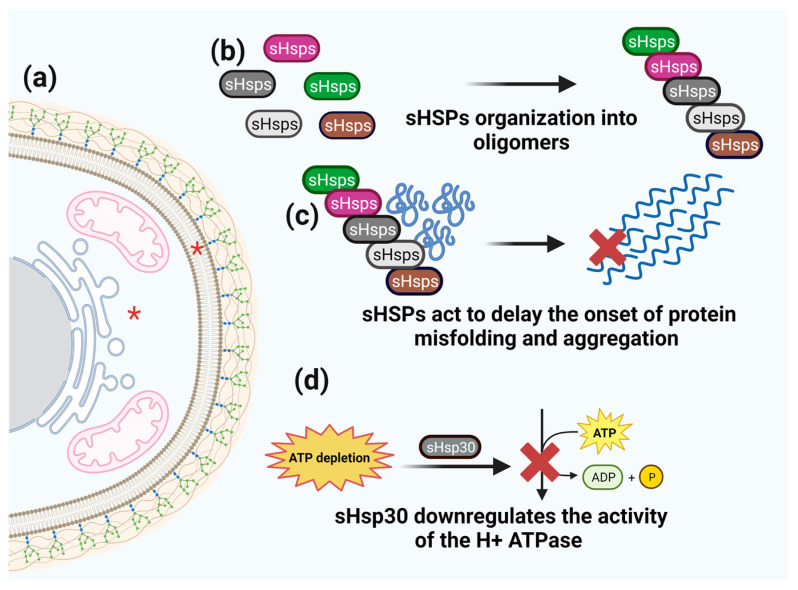
**Schematic representation of sHsps function.** (**a**) sHSPs cell localization is marked by red asterisks. (**b**) sHSPs organize into oligomers. (**c**) Oligomers act to delay of the onset of protein misfolding and aggregation. (**d**) sHsp30 downregulates the activity of the H+ ATPase under ATP starvation.

**Figure 3 microorganisms-11-01878-f003:**
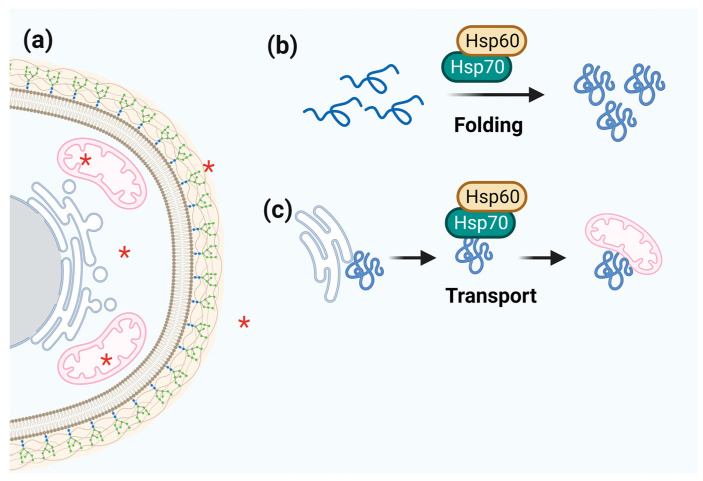
**Schematic representation of Hsp60 function.** (**a**) Hsp60 cell localization is marked by red asterisks. (**b**,**c**) Hsp60 acts together with Hsp70 to fold and transport target proteins.

**Figure 4 microorganisms-11-01878-f004:**
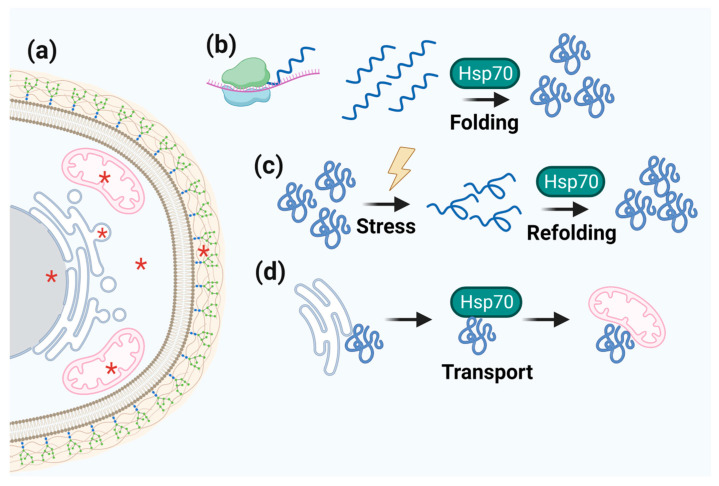
**Schematic representation of Hsp70 function.** (**a**) Hsp70 cell localization is marked by red asterisks. (**b**–**d**) The centralized role of Hsp70 in folding, refolding, and transporting client proteins.

**Figure 5 microorganisms-11-01878-f005:**
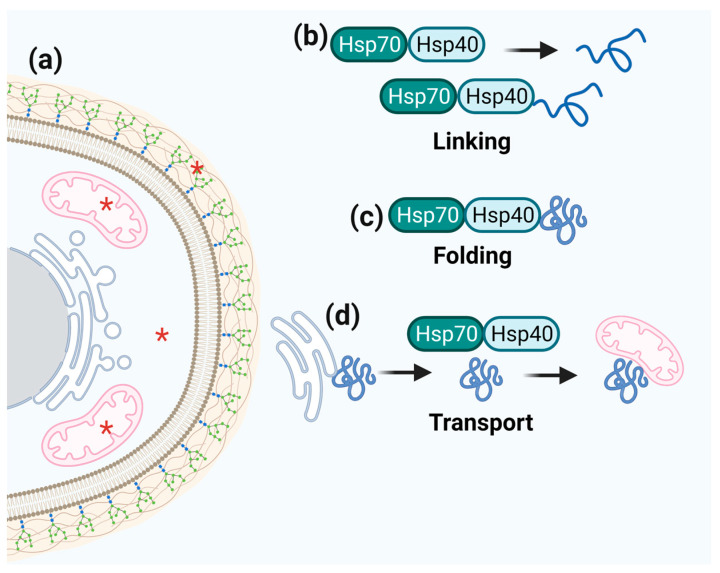
**Schematic representation of Hsp40 function.** (**a**) Hsp40 cell localization is marked by red asterisks. (**b**–**d**) Hsp40 links to Hsp70 to fold and transport numerous target proteins that participate in a wide range of cellular processes.

**Figure 6 microorganisms-11-01878-f006:**
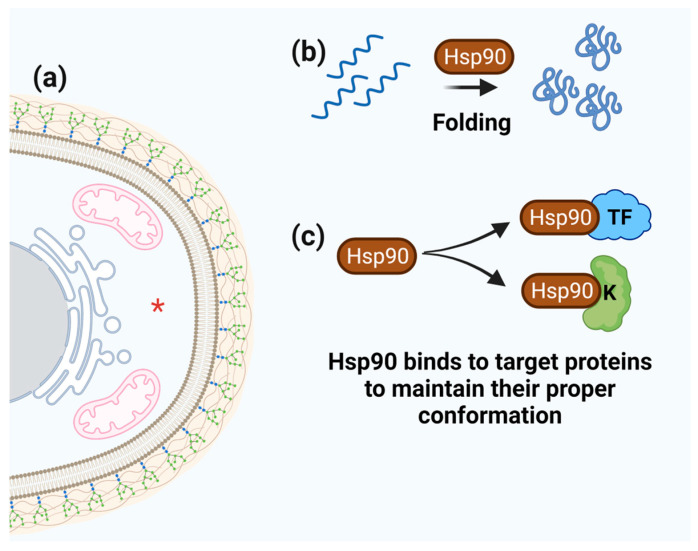
**Schematic representation of Hsp90 function.** (**a**) Hsp90 cell localization is marked by red asterisk. (**b**) The ability of Hsp90 to fold numerous target proteins. (**c**) Hsp90 binds and maintains the proper conformation of target proteins (e.g., transcription factors and protein kinases).

**Figure 7 microorganisms-11-01878-f007:**
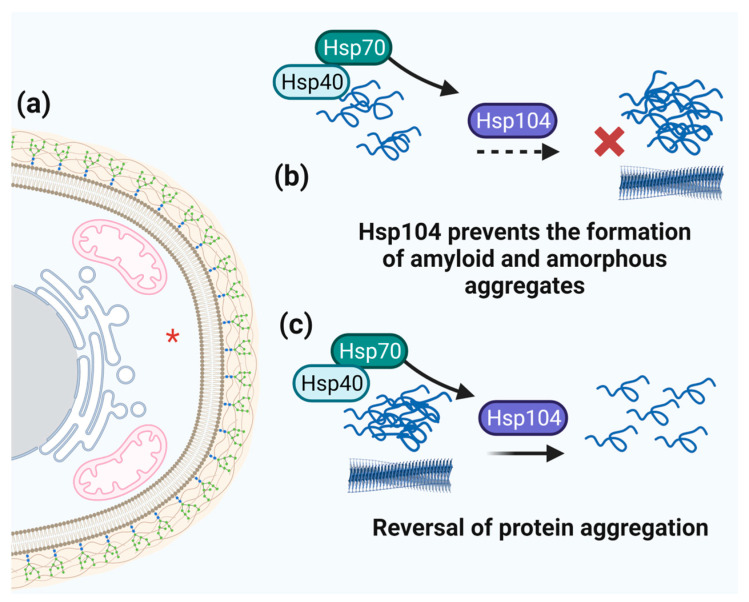
**Schematic representation of Hsp104 function.** (**a**) Hsp104 cell localization is marked by red asterisk. (**b**) The ability of Hsp104 to prevent protein aggregation (assisted by Hsp70 and Hsp40 directing client proteins to Hsp104). (**c**) The role of Hsp104 in reversing amorphous and amyloid protein aggregates.

**Figure 8 microorganisms-11-01878-f008:**
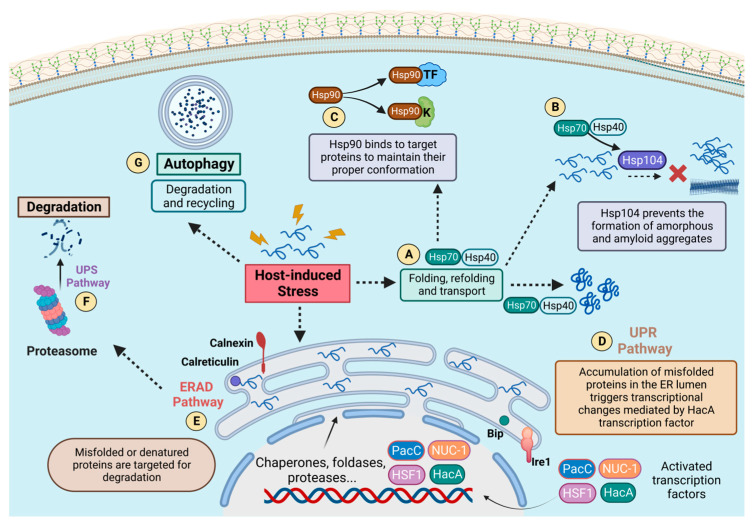
**Pathways and HSPs recruited for cellular proteostasis in response to stress.** The functionality of the different pathways is represented. Such molecular circuits support cellular proteostasis either by monitoring the conformation of endoplasmic reticulum proteins and activating necessary transcriptional changes through the action of transcriptional factors (UPR—(**D**)), targeting misfolded proteins for proteasomal degradation (ERAD—(**E**) and UPS—(**F**)), or degrading and recycling proteins via autophagy by lysosomes (Autophagy—(**G**)). The major HSPs are also represented. Hsp70 performs a centralized role in the cell in association with other HSPs, such as Hsp40 (**A**) and Hsp104 (**B**), acting for folding, refolding, and transport functions, as well as preventing the formation of amorphous and amyloid aggregates. Hsp90 is also represented (**C**) and plays a critical role in stress-responsive circuits, maintaining the conformation of very specific client proteins, such as kinases and transcription factors. These mechanisms work together to ensure cellular proteostasis, enabling the adaptation and resistance of fungal pathogens under stress conditions induced by the host environment.

**Table 1 microorganisms-11-01878-t001:** Proteostasis circuits and their functions, cell localization, and biological processes related to fungal pathogenicity. Drugs targeting HSPs and other proteostasis components are also shown.

Proteostasis Components	Function	Cellular Location	Biological Process Involved in Fungi	Targeting Drugs	References
Small HSPs	Prevent the onset of protein misfolding and aggregation and have the ability to form HSP oligomers.	Cytosol close to and on the cell membrane.	Stress response following nucleotide depletion or DNA damage; tolerance to thermal/cold, osmotic, and oxidative stress; phagocytosis and virulence.	Apatorsen (OGX-427); Quercetin (Qctn).	[96,98,99,101,102,165]
Hsp60	Auxiliary to Hsp70 in the folding and transport of client proteins through the cell.	Especially in mitochondria but also in the cytosol, the cell wall, and the extracellular space.	Ligands of CD11/CD18 of human macrophages; immunodominant antigens resulting in humoral and cellular responses.	Mefipristone (RU486); Pyrazolopirimidine (EC3016); Mizoribine; Avrainvillamide; Epolactaene; Suvanine.	[65,117,118,119,166]
Hsp70	Centralized function in proteostasis circuits. Folding and assembly of newly synthesized proteins, refolding of denatured or aggregated proteins, protein transport, and degradation mediated by ERAD and the UPR.	All major cell compartments: cytosol, nucleus, ER, and mitochondria.	Binding to human salivary histatin 5; germination, conidiation, and sporulation; interaction with the MAPK pathway.	Pifithrin-µ (2-PES); KNK437; Methylene Blue; Imidazole derivates (apoptozole); Pyrrhoricin, Oncocin and derivates; KLR-70; Adenosine derivates (VER-155008); S1g-2 andS1g-6; Benzimidazole class; 2,5′-thiodipyrimidine (YK-5).	[123,127,128,130,167,168]
Hsp40	Linker between the substrate and Hsp70; Pairs with Hsp70 to promote protein refolding and degradation.	Cytosol, cell wall, and mitochondria.	Host–pathogen interaction, growth, and yeast–mycelium transition.	KNK437; Quercetin (Qcnt); Plumbagin derivates (PLIHZ and PLTFBH).	[80,123,131,132]
Hsp90	Controls the activity of regulatory proteins such as transcription factors and kinases, all of which participate in a wide range of cellular processes.	Cytosol.	Regulation of the cell cycle and yeast-to-hyphae transition; hyphal growth, reproduction, and development at high temperatures (dependent on Ras1 signaling system); biofilms and host cell attachment; colonization of keratinized tissues.	Benzoquinone Ansamycin class (Geldanamycin, 17-AAG, 17-DMAG, Ganetespib); Trichostatin A; Enfungumab; Radicicol; Novobiocin derivates; Deguelin.	[80,136,169]
Hsp104	Prevention and reversal of protein aggregation assisted by Hsp70 and Hsp40.	Cytosol.	Biofilm development; crosstalk with the autophagy mechanism.	Suramin; Guanidine Hydrochloride; NSC 34931; NSC 71948; Hexachlorophene.	[154,157,158]
UPR Pathway	Triggered by ER stress caused by the accumulation of toxic misfolded proteins; upregulation of genes that support ER function to rebalance proteostasis.	Endoplasmic Reticulum.	Response and adaptation to environmental stress; virulence; nutrient adaptation and host invasion; keratinolytic capacity; thermotolerance; growth in host molecules and resistance to antifungals.	**GRP78**: Hyfroxyquinolines; Epigallocatechin gallate (EGCG), and Honokiol (HNK) **IRE1**: Kinase-biding RNase-attenuators (KIRAs); MKC analogues, and salicylaldehydes **ERAD**: Eeyarestatin I **ER Stress**: Dithiothreitol (DTT) and Tunicamycin.	[67,69,170,171,172,173]
Autophagy	Recycling of intracellular components.	Autophagosomes.	Mechanism to cope with nutrient deprivation; related to the expression of HSPs.	**VPS34**: PIK-III **mTOR**: AZD8055 **Lysosomal Function**: Bafilomycin A.	[5,33,34,35,36,37]
Proteasome	Multicatalytic complexes that cleave proteins into peptides.	Cytosol and nucleus.	F-box proteins from the UPS are related to the following: pseudo-hyphal production, dimorphic switch, cell membrane integrity, sexual sporulation, carbon utilization, signal transduction, and nutrient sensing.	Bortezomib (Velcade); Carfilzomib (Kyprolis); Ixazomib (Ninlaro); Mg-132; Lactacystin.	[174,175,176]
HSF1	A transcription factor that binds to heat shock elements (HSE) in the promoter regions of HSP and related genes, activating the heat shock response (HSR).	Cytosol and nucleus.	Virulence; cell wall integrity; stress response.	**Natural Inhibitors**: Quercetin, Stresgenin B, Triplotide; Cantharidin; Fisetin; Rocaglamide A; CL-43; 2,4-Bis(4-hydroxybenzyl)Phenol. **Synthetic Inhibitors**: KNK437; NZ-28; KRIBB11; PW3405; IHSF115; 4,-6-Disubstituted Pyrimidine; CCT251236; α-Acyl Amino Carboxamides.	[56,177,178,179]

## Data Availability

No new data were created or analyzed in this study. Data sharing is not applicable to this article.

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
