# Peer review of "Insights and Perspectives on the Role of Proteostasis and Heat Shock Proteins in Fungal Infections"

_microorganisms, 2023, doi:10.3390/microorganisms11081878_

Round 1
Reviewer 1 Report
The manuscript entitled „Insights and Perspectives on the Role of the Proteostasis and Heat Shock Proteins in Fungi Infections” is about the role of fungal HSPs in the infection of the host. This is an interesting issue due to increasing cases of multidrug-resistant fungal infection.
In the beginning, the manuscript concentrated on the role of protein proteostasis during infection. Then, the authors describe the HSPs as separate groups according to molecular weight. The authors concentrated on understanding the role of fungal HSPs in infection. I think that the section about the host (human) HSPs in fungal infection should be included. It is desirable to include the section about the homology of fungal and human HSPs and how they can affect each other. May the high homologous of HSPs between fungal and human HSP mask the response of the immune system?
The authors also summarize the information about the HSP in Table 1. I wonder whether the references and targeting drugs are relevant for fungal HSPs. Several references correspond to human HSPs and their role in diseases.
In the line 51, there is "HSR is regulated by heat shock proteins.." This is not correct. The HSR is regulated by HSF1. HSR is mediated by by HSP. Please, correct.
Author Response
The manuscript entitled “Insights and Perspectives on the Role of the Proteostasis and Heat Shock Proteins in Fungi Infections” is about the role of fungal HSPs in the infection of the host. This is an interesting issue due to increasing cases of multidrug-resistant fungal infection.
In the beginning, the manuscript concentrated on the role of protein proteostasis during infection. Then, the authors describe the HSPs as separate groups according to molecular weight. The authors concentrated on understanding the role of fungal HSPs in infection. I think that the section about the host (human) HSPs in fungal infection should be included. It is desirable to include the section about the homology of fungal and human HSPs and how they can affect each other. May the high homologous of HSPs between fungal and human HSP mask the response of the immune system?
Reply: Thanks for the suggestion. Including a section about hosts' HSPs is outside our review's scope. There is so much information about plants, animals, and human HSPs that would mischaracterize the manuscript. Although we have discussed some HSPs of fungi that infect plants and animals, we are not experts in the HSPs of animals and plants.
The authors also summarize the information about the HSP in Table 1. I wonder whether the references and targeting drugs are relevant for fungal HSPs. Several references correspond to human HSPs and their role in diseases.
Reply: We added a sentence with information on drugs targeting those components that might be helpful to guide further studies in proteostasis and fungi. That should be true regardless of the model organism used in the references to characterize the drugs.
In the line 51, there is "HSR is regulated by heat shock proteins." This is not correct. The HSR is regulated by HSF1. HSR is mediated by HSP. Please, correct.
Reply: Done as requested. We fixed the sentence accordingly.
Reviewer 2 Report
This manuscript by Neves-da-Rocha et al is a review of the roles of heat shock proteins and other processes for intra-cellular protein degradation in fungal pathogens. The paper is a tour de force. It is clearly written and nicely illustrated. I have very few comments:
1. They emphasize work on Paracocci and other primary pathogenic dimorphic fungi are barely mentioned. In particular, Histoplasmosis is left out, although starting with Kobayashi there has been a fair amount of work correlating metabolic and proteomic changes with shift from mold to yeast (Cleare LG, Zamith-Miranda D, Nosanchuk JD. Heat Shock Proteins in Histoplasma and Paracoccidioides. Clin Vaccine Immunol. 2017 Nov 6;24(11):e00221-17. doi: 10.1128/CVI.00221-17. PMID: 28903987; PMCID: PMC5674194.)
2. Table 1 is a very interesting compilation of HSPs and “target drugs”. The problem is that HSPs are conserved across kingdoms, as is calcineurin. FK506 is very immunosuppressive so not a real consideration as an antifungal. Apropos of calcineurin, cyclosporine A also has antifungal activity against both Cryptococcus and Coccidioides.
3. Line 227- What is meant by average temperatures?
Author Response
This manuscript by Neves-da-Rocha et al is a review of the roles of heat shock proteins and other processes for intra-cellular protein degradation in fungal pathogens. The paper is a tour de force. It is clearly written and nicely illustrated. I have very few comments:
- They emphasize work on Paracocci and other primary pathogenic dimorphic fungi are barely mentioned. In particular, Histoplasmosis is left out, although starting with Kobayashi there has been a fair amount of work correlating metabolic and proteomic changes with shift from mold to yeast (Cleare LG, Zamith-Miranda D, Nosanchuk JD. Heat Shock Proteins in Histoplasma and Paracoccidioides. Clin Vaccine Immunol. 2017 Nov 6;24(11):e00221-17. doi: 10.1128/CVI.00221-17. PMID: 28903987; PMCID: PMC5674194.).
Reply: We added additional information on dimorphic fungi, specifically for Histoplasma.
- Table 1 is a very interesting compilation of HSPs and “target drugs”. The problem is that HSPs are conserved across kingdoms, as is calcineurin. FK506 is very immunosuppressive so not a real consideration as an antifungal. Apropos of calcineurin, cyclosporine A also has antifungal activity against both Cryptococcus and Coccidioides.
Reply: We clarified this point by mentioning the limitations of FK506 and suggesting cyclosporine A as an alternative.
- Line 227- What is meant by average temperatures?
Reply: We changed that to "ambient temperature".
Reviewer 3 Report
Fungal infection become more significant because of the increase of immunocompromised population. This review summarizes the roles of proteostasis and heat-shock proteins in fungal infections and includes a large amount information about the progress in this field. Recent studies indicate that proteostasis and heat-shock proteins are promising targets for antifungal therapies especially in combination with other antifungal drugs. Some concerns need to be addressed before publication.
One major issue is that this review covers so many different fungal species and proteins/genes, thus it is not easy for readers to follow. One way to address this issue is to include a diagram to give readers a summary of different groups of HSPs at the beginning of section 3 (Heat Shock Proteins). In addition, some information is missing for proteins and drugs, which causes confusion for readers who are not very familiar with this field. Below are the details.
Page 2: A brief explanation is needed for the PacC/RIM101 factor.
Page 3: A brief introduction of NUC-1, cyclin-CDK, Hsf1 is missing. The evidence of the function of Hsp70/90 in oxidative response and proteolysis is not well presented.
Page 9: The sentence “In A. fumigatus, thermotolerance increases from 25 to 35 °C” is incomplete.
Page 12: More explanation about the molecular transistor is needed. It is not clear how Hsp90 allows client kinases to accumulate point mutations. Are these mutations at protein or DNA level?
Page 13: It is better to show the evidence for the crosstalk between Hsp104 and autophagy machinery. Also, the roles of Hsp104 in protein-based epigenetics and biofilm are not clearly explained.
Page 15: A brief introduction about the mechanism of action of Caspofungin, Amphotericin B, and azole drugs will help readers for a better understanding of the context.
Page 16: It is unclear how inhibition of Hsp90 activity blocks the emergence of resistant phenotypes. The following sentence needs to be reworded. “These inhibitors were designed to specifically target the ATPase activity of Hsp104, disrupting its function and preventing the disaggregation of protein aggregates”.
Minor edition is required for this review.
Author Response
Fungal infection become more significant because of the increase of immunocompromised population. This review summarizes the roles of proteostasis and heat-shock proteins in fungal infections and includes a large amount information about the progress in this field. Recent studies indicate that proteostasis and heat-shock proteins are promising targets for antifungal therapies especially in combination with other antifungal drugs. Some concerns need to be addressed before publication.
One major issue is that this review covers so many different fungal species and proteins/genes, thus it is not easy for readers to follow. One way to address this issue is to include a diagram to give readers a summary of different groups of HSPs at the beginning of section 3 (Heat Shock Proteins). In addition, some information is missing for proteins and drugs, which causes confusion for readers who are not very familiar with this field. Below are the details.
Page 2: A brief explanation is needed for the PacC/RIM101 factor.
Reply: We have explained in the text, with references, the mechanism for the PacC activation and function.
Page 3: A brief introduction of NUC-1, cyclin-CDK, Hsf1 is missing. The evidence of the function of Hsp70/90 in oxidative response and proteolysis is not well presented.
Reply: We have explained in the text, with references, the mechanism and function of the fungal Pi acquisition system and the role of the Heat Shock Factor. We added a new sentence concerning HSPs and oxidative stress, although that is a common sense understanding in the field.
Page 9: The sentence “In A. fumigatus, thermotolerance increases from 25 to 35 °C” is incomplete.
Reply: We completed the sentence; thank you.
Page 12: More explanation about the molecular transistor is needed. It is not clear how Hsp90 allows client kinases to accumulate point mutations. Are these mutations at protein or DNA level?
Reply: We changed a lot in this paragraph, adding more information on the role of Hsp90. Notedly, there are two particular references for this understanding.
Page 13: It is better to show the evidence for the crosstalk between Hsp104 and autophagy machinery. Also, the roles of Hsp104 in protein-based epigenetics and biofilm are not clearly explained.
Reply: We added more information to those passages for a better understanding; thank you.
Page 15: A brief introduction about the mechanism of action of Caspofungin, Amphotericin B, and azole drugs will help readers for a better understanding of the context.
Reply: We added the mechanisms of action for these 3 drugs.
Page 16: It is unclear how inhibition of Hsp90 activity blocks the emergence of resistant phenotypes. The following sentence needs to be reworded. “These inhibitors were designed to specifically target the ATPase activity of Hsp104, disrupting its function and preventing the disaggregation of protein aggregates”.
Reply: We reworded this sentence. As for Hsp90, we added a reference in this passage, and the previous clarification of its transistor function should also solve this issue.
Round 2
Reviewer 1 Report
Thank you for your reply. I am satisfied with it.